# Is Poor Lithium Response in Individuals with Bipolar Disorder Associated with Increased Degradation of Tryptophan along the Kynurenine Pathway? Results of an Exploratory Study

**DOI:** 10.3390/jcm11092517

**Published:** 2022-04-29

**Authors:** Frederike T. Fellendorf, Mirko Manchia, Alessio Squassina, Claudia Pisanu, Stefano Dall’Acqua, Stefania Sut, Sofia Nasini, Donatella Congiu, Eva Z. Reininghaus, Mario Garzilli, Beatrice Guiso, Federico Suprani, Pasquale Paribello, Vittoria Pulcinelli, Maria Novella Iaselli, Ilaria Pinna, Giulia Somaini, Laura Arru, Carolina Corrias, Federica Pinna, Bernardo Carpiniello, Stefano Comai

**Affiliations:** 1Section of Psychiatry, Department of Medical Sciences and Public Health, University of Cagliari, 09121 Cagliari, Italy; frederike.fellendorf@medunigraz.at (F.T.F.); m.garzi@gmail.com (M.G.); beatrice.guiso@gmail.com (B.G.); federicosuprani@hotmail.it (F.S.); pasqualeparibello@gmail.com (P.P.); vittoriapulcinelli@hotmail.com (V.P.); novella.iaselli@gmail.com (M.N.I.); ilaria.pinna1991@gmail.com (I.P.); giulia444@alice.it (G.S.); laura.arru282@gmail.com (L.A.); carol.corrias@gmail.com (C.C.); fedepinna@inwind.it (F.P.); bcarpini@iol.it (B.C.); 2Psychiatry and Psychotherapeutic Medicine, Medical University Graz, 8010 Graz, Austria; eva.reininghaus@medunigraz.at; 3Department of Pharmacology, Dalhousie University, Halifax, NS B3H 0A2, Canada; 4Unit of Clinical Psychiatry, University Hospital Agency of Cagliari, 09121 Cagliari, Italy; 5Department of Biomedical Science, Section of Neuroscience and Clinical Pharmacology, University of Cagliari, Monserrato, 09042 Cagliari, Italy; squassina@unica.it (A.S.); claudia.pisanu@unica.it (C.P.); dcongiu@unica.it (D.C.); 6Department of Pharmaceutical and Pharmacological Sciences, University of Padova, 35131 Padova, Italy; stefano.dallacqua@unipd.it (S.D.); stefania.sut@unipd.it (S.S.); sofia.nasini@studenti.unipd.it (S.N.); stefano.comai@unipd.it (S.C.); 7Department of Biomedical Sciences, University of Padova, 35131 Padova, Italy; 8San Raffaele Scientific Institute, 20132 Milano, Italy; 9Department of Psychiatry, McGill University, Montreal, QC H3A 1A1, Canada

**Keywords:** bipolar disorder, tryptophan, kynurenine, indoleamine 2,3 dioxygenase (IDO), lithium response, Alda scale

## Abstract

Bipolar disorder is associated with an inflammation-triggered elevated catabolism of tryptophan to the kynurenine pathway, which impacts psychiatric symptoms and outcomes. The data indicate that lithium exerts anti-inflammatory effects by inhibiting indoleamine-2,3-dioxygenase (IDO)-1 activity. This exploratory study aimed to investigate the tryptophan catabolism in individuals with bipolar disorder (*n* = 48) compared to healthy controls (*n* = 48), and the associations with the response to mood stabilizers such as lithium, valproate, or lamotrigine rated with the Retrospective Assessment of the Lithium Response Phenotype Scale (or the Alda scale). The results demonstrate an association of a poorer response to lithium with higher levels of kynurenine, kynurenine/tryptophan ratio as a proxy for IDO-1 activity, as well as quinolinic acid, which, overall, indicates a pro-inflammatory state with a higher degradation of tryptophan towards the neurotoxic branch. The treatment response to valproate and lamotrigine was not associated with the levels of the tryptophan metabolites. These findings support the anti-inflammatory properties of lithium. Furthermore, since quinolinic acid has neurotoxic features via the glutamatergic pathway, they also strengthen the assumption that the clinical drug response might be associated with biochemical processes. The relationship between the lithium response and the measurements of the tryptophan to the kynurenine pathway is of clinical relevance and may potentially bring advantages towards a personalized medicine approach to bipolar disorder that allows for the selection of the most effective mood-stabilizing drug.

## 1. Introduction

Bipolar disorder (BD) is a severe and lifelong mental disorder that is characterized by recurrent depressive and manic or hypomanic episodes that alternate with times without illness symptoms [1]. The psychopharmacology of BD is still a matter of research, since up to 20 to 50 percent of treated patients do not respond sufficiently to maintenance therapies [2]. In this context, the identification of biomarkers that provide information about the current stage of illness and the expected pharmacological response is a key step toward a personalized approach to BD [3].

Tryptophan (TRP) is an essential amino acid that is relevant to protein synthesis and that, through its metabolites, is also relevant to mood, cognition, and sleep. TRP has two major biochemical catalytic pathways: the serotonin (5-HT) and the kynurenine (KYN) pathways, which are illustrated in Figure 1 [4]. Via the enzyme tryptophan 5-hydroxylase, TRP is converted into 5-hydroxytryptophan, which is then transformed into 5-HT by the 5-hydroxytryptophan decarboxylase enzyme. 5-HT can be degraded into 5-hydroxyindoleacetic acid, or it can be transformed into melatonin (MLT) at the levels of the pineal gland. The other metabolic route of TRP is the KYN pathway, which depends on the activity of the first two rate-limiting enzymes: the tryptophan 2,3-dioxigenase (TDO) and the indoleamine 2,3-dioxygenases (IDO)-1 and 2 [5,6]. KYN is further catabolized by the enzymes kynurenine-aminotransferase (KAT) into kynurenic acid (KYNA), or by the kynurenine-3-monooxygenase (KMO) into 3-hydroxykynurenine (3-HK). 3-HK is then converted into 3-hydroxyanthranilic acid (3-HAA) by the enzyme kynureninase, and the 3-HAA is finally degraded non-enzymatically into quinolinic acid (QA) [4].

The tryptophan degradation is affected by sex [7], age [8], smoking [9], weight [10], and by the somatic as well as the mental health status. KYNA has neuroprotective affects that act as antagonists at the n-methyl-d-aspartate-(NMDA) receptor, while QA has neurotoxic and free-radical-producing properties that are due to its agonistic effects at the NMDA receptors [4,11,12], and that also induce apoptosis in astrocytes [13]. Therefore, the catabolism of TRP is assumed to impact the genesis, symptoms, and outcomes of neuropsychiatric disorders [4,14,15], including BD. Indeed, BD is associated with immunological and low-grade inflammation processes with elevated pro-inflammatory cytokines and acute-phase proteins [16]. Increased IDO-1 activity is assumed under pro-inflammatory conditions [17], as it is induced by pro-inflammatory cytokines (mainly interferon-γ, but also interleukin (IL)-2 and -6 and tumor necrosis factor-α) and it is inhibited by anti-inflammatory cytokines such as IL-4 [4,18,19,20]. In contrast, inflammatory processes might reduce KAT activity [21]. Therefore, an increased TRP degradation towards the KYN pathway in BD has been discussed [22,23,24,25,26] as a factor that could impact mood symptoms [27]. The elevated catabolism of serum TRP to KYN may lead to a reduction in the circulating TRP that is available for the production of 5-HT and MLT [4,17], which affect the psychiatric symptomatology (e.g., mood and circadian rhythms, as well as comorbidities and drug response [28,29]). Alternatively, central 5-HT synthesis might also be independent of peripheral TRP catabolites [30].

Pharmacological maintenance treatment to prevent relapses is highly recommended in BD [2], with a high level of evidence for lithium [31], anticonvulsants such as valproate and lamotrigine, as well as some second-generation antipsychotics, including quetiapine, olanzapine, aripiprazole, and asenapine [2]. The research is scant about the effects of these drugs on the metabolism of TRP, and about the possible relationship between TRP metabolism and the treatment response to these drugs. However, the data show that lithium has anti-inflammatory effects through the inhibition of IDO activity [32]. Furthermore, lithium seems to be associated with increased MLT secretion [33]. Additionally, a trial found a decreased sensitivity of MLT to light that was related to valproate intake in healthy humans [34]. In individuals with unipolar depression, the TRP metabolites along the KYN pathway were shown to be potential biomarkers of the antidepressant treatment response [35,36].

In this not-yet-clear and novel context, we took advantage of our ongoing study that is aimed at assessing the relationship between BD and tryptophan catabolism along the 5-HT to the MLT and KYN pathways, and the mood-stabilizer response [37]. After analyzing whether peripheral levels of TRP and its metabolites via 5-HT and KYN are altered in euthymic individuals with BD compared to healthy controls (HC), we investigated the plasma levels of TRP and its metabolites in relation to the response to the pharmacotherapy with mood stabilizers, including lithium, valproate, or lamotrigine. We hypothesized that: (1) Individuals with BD show a greater degradation of TRP along the KYN pathway, with the increased formation of neurotoxic over neuroprotective metabolites; and (2) The mood stabilizers lithium and valproate, with anti-inflammatory properties, are associated with a reduced degradation of TRP along the KYN pathway.

## 2. Materials and Methods

### 2.1. Setting and Participants

Participants were recruited consecutively at the Psychiatric Unit of the University Hospital of Cagliari and of the Department of Medical Sciences and Public Health, the University of Cagliari. Briefly, the study is part of an ongoing project that is aimed at exploring the relation between BD, MLT, gut microbiota, and genetics in a longitudinal setting with euthymic as well as depressive and manic states in a cohort of 50 BD and 50 HC patients [37]. This sample size was deemed adequately powered (more than 95%) to detect a difference between groups, with an α set at 0.05, and considering an effect size in the difference of the MLT levels between HC and BD patients equal to 5.3 (see [38] for further details). For the current study, only the cross-sectional data from 48 individuals with BD in the euthymic phase as well as 48 HC were analysed. Indeed, in the BD group, two participants were not in a euthymic state. To exclude this possible confounder, we included only euthymic participants. In the control group, one participant was excluded because of missing sociodemographic data, and one because of missing laboratory data. Diagnoses and clinical assessments were performed by trained psychiatrists through direct interview and through a systematic review of the patients’ medical records. The diagnosis of BD was confirmed with the Italian version of the Structured Clinical Interview (SCID), according to the Diagnostic and Statistical Manual of Mental Disorders (DSM)-5 [38]. The following inclusion criteria were applied for the population of patients with BD: (a) The presence of a diagnosis of BD type 1 or type 2, according to the DSM-5 criteria; and (b) Aged between 18 and 65 years old. Individuals were excluded if: (a) They were women at the fertile age who did not use adequate contraception or who were pregnant; (b) They had a history of traumatic brain insults; (c) They had a diagnosis of current and/or a lifetime of other psychiatric or neurological disorders, or other severe unregulated medical conditions; (d) They had a diagnosis of current and/or a lifetime of substance-use disorder; (e) They had been treated with melatonergic compounds (melatonin and/or agomelatine) for at least two months before enrolment. Other treatments (beta blockers, benzodiazepines, low-dose antipsychotics) were allowed. The inclusion criteria for the HC were: (a) The absence of psychiatric disorders diagnosed according to the DSM-5; (b) Aged between 18 and 65 years old; and (c) The absence of psychiatric, neurological, or other severe unregulated medical conditions. The HC were recruited by word of mouth among the hospital staff, their families, and university students. They underwent a standard medical and laboratory test assessment to verify their health status, and they were clinically assessed to establish the absence of any previous and current mental disorders. Furthermore, the cases and the HC were matched for age and sex. The participants provided written informed consent before they participated in this study. The assessment included sociodemographic parameters; the illness history, including the number and entity of illness episodes; psychotic features and suicide attempts; the medication history; the current psychiatric symptomatology, measured with validated rating scales; the treatment response using the *Retrospective Criteria of Long-Term Treatment Response in Research Subjects with Bipolar Disorder* (Alda scale) scale; the body mass index (BMI); and the fasting blood between 8.00 a.m. and 10.00 a.m. The study was approved by the local ethics committee (Ethics Committee of the University Hospital Agency of Cagliari: PG/2019/6277) in compliance with the current revision of the Declaration of Helsinki and the current EU regulations for the protection of privacy.

### 2.2. Characterization of Response to Treatment and Psychometric Assessment

In this study, individuals with BD taking either lithium, valproate, lamotrigine, or any combination of these were rated according to this scale. Specifically, we performed a detailed revision of the clinical charts to permit the graphic depiction of the longitudinal clinical course with the life-chart method. In our study, this included both the retrospective assessment of the past clinical course (based on accurate longitudinally collected clinical data) and the prospective two-year observation period. Then, we calculated the area under the curve-of-illness activity (severity of the episodes x duration of episodes) before and after the introduction of a mood stabilizer or a combination of mood stabilizers. This provided us with an objective measure of the clinical improvement under a specific treatment or combination of treatments (Criterion A of the scale). Indeed, the Alda scale was developed to evaluate the response treatment with mood stabilizers retrospectively [39,40]. It consists of two main criteria: Criterion A expresses the clinical improvement under a specific treatment. This score is then penalized by five criteria (B): (B1) The number of episodes before/off the treatment; (B2) The frequency of episodes before/off the treatment; (B3) The duration of the treatment; (B4) The compliance during the period(s) of stability; and (B5) The use of additional medication during the period of stability. The total score is obtained by subtracting the sum of the B criteria from that of Criterion A, and it ranks between 0 and 10 points. A total score ≥ 7 is classified as a good responder, while a total score < 7 includes partial and nonresponders [39,40,41]. Importantly, the Alda scale was validated for the assessment of the response to other mood stabilizers, such as valproic acid, lamotrigine, carbamazepine, and atypical antipsychotics, as well as combination therapies [42,43]. The assessment of the clinical response to mood stabilizers was performed by trained psychiatrists, under the supervision of one senior rater (M.M.) who has worked in the validation procedure of the scale [40].

In this naturalistic study, some patients took combinations of mood stabilizers for variable durations, or they were treated sequentially with diverse mood stabilizers. Here, we are expressing the improvement that was observed under each mood stabilizer, and not the concomitant treatment.

The *Hamilton Rating Scale for Depression* (21 items) (HAMD) [44] is an external rating scale with the aim of determining the severity of depressive symptoms in patients who were already diagnosed with depression. The items are rated with two or four points. In the current version, up to 65 points can be awarded. A cut-off value was not originally planned, and therefore various values were used to classify the euthymia and subsyndromal groups over the years. The *Young Mania Rating Scale* (YMRS) [45] is an external rating scale that is used to determine the severity of manic symptoms in patients with BD. The YMRS consists of eleven items, with five levels of severity. Euthymia was defined as a HAMD score < 14 and a YMRS score < 9 points. Current anxiety was measured with the *Hamilton Rating Scale for Anxiety* (21 items) (HAMA) [46]. The Clinical Global Impression-Severity (CGI) [47] was used to assess the psychopathology and the functioning, with a short external rating on a 1–7 rating scale.

### 2.3. Laboratory Methods

The blood samples were collected in EDTA-containing tubes that were immediately centrifuged at 2500 rpm at 4 °C for 10 min. Plasma was then aliquoted and stored at −80 °C until the analysis. The analysis of the plasma levels of TRP and its metabolites along the 5-HT and the KYN pathways was conducted within 4 months from the collection, according to the standard methods in our lab at the University of Padova [20,48,49]. The TRP, 5-hydroxytryptophan, 5-HT, and KYN were determined by using an HPLC system equipped with a UV–Vis and fluorometric detectors, whereas the QA, KYNA, 3-HK, and MLT were determined by LC-MS/MS by using alfa-methyltryptophan as an internal standard. Details on the methodology can be found in our recent papers [20,48,49].

### 2.4. Statistical Analyses

All analyses were performed with the IBM Statistical Package for Social Sciences (SPSS), version 27.0. After checking for the normal distribution of the data by using the Kolmogorov–Smirnov test, differences in the demographic data between the BD and HC groups were tested with either a chi-square test (sex, smoking status, nominal data) or unpaired *t*-tests (metric data, normally distributed according to the Kolmogorov–Smirnov test). To analyze the differences between the groups for the plasma levels of TRP and its catabolites via 5-HT and KYN, a multivariate analysis of variance (MANCOVA) with the covariates sex, age, BMI, and smoking (yes, no, ex-smoker) and the Bonferroni correction was conducted. Unpaired *t*-tests and Mann–Whitney U tests (metric data, not normally distributed) were used to analyze the difference between responders and non-responders according to the Alda-scale score in all the catabolite levels. Pearson (normally distributed data) and Spearman (not normally distributed data) correlation coefficients were calculated to analyze the relations of the catabolites with the Alda-scale score. The statistical significance threshold was set at *p* < 0.05.

## 3. Results

### 3.1. Comparison of Individuals with Bipolar Disorder and Healthy Controls

Table 1 presents the demographic data and the concentrations of the TRP catabolites of individuals with BD and HC. The group effect of the MANOVA showed significant differences between the groups in the catabolites (*F* (10,82) = 4.668; *p* < 0.001; *η*^2^ = 0.363). Age, sex, BMI, and smoking did not show significant impacts. The post-hoc analyses found lower TRP, lower QA, and higher 5-HTP in euthymic BD than in HC. The patients had been treated with lithium for an average of 10 years, with a standard deviation of ±9.2 years.

### 3.2. Relationship of Treatment Response to Tryptophan Catabolites

The illness-specific data about current symptomatology, the number of episodes, and the current pharmacological mood-stabilizing treatments of the participants with BD are reported in Table 2. Almost all the BD participants were treated with mood-stabilizing medication, with around half of them taking a monotherapy with lithium, anticonvulsants, or antipsychotics, and with the other half taking a combination therapy (see Table 2). Four participants were not treated with sufficient mood stabilizers and were therefore not included in the correlation analyses. Two patients were treated with carbamazepine and lithium, but only the lithium response was considered. The three patients taking antipsychotics, topiramate, or carbamazepine, respectively, or only oxcarbazepine, were not assessed for response. The total lithium Alda score of the participants taking lithium as mono- or as combination therapy negatively correlated with KYN (*r* (15) = −0.49, *p* = 0.047), KYN/TRP (*r* (15) = −0.52, *p* = 0.035), and QA (*r* (15) = −0.61, *p* = 0.009) (Figure 2). Interestingly, only in the participants taking lithium monotherapy was the negative relationship between the Alda-scale score and QA (*r* (6) = −0.88, *p* < 0.004) found. Conversely, the associations of the Alda-scale score and KYN (*r* (7) = −0.70, *p* = 0.037) as well as KYN/TRP ratio (*r* (7) = −0.78, *p* = 0.013) were present only in the individuals taking a combination with either an anticonvulsant or an antipsychotic. The Alda-scale score of the individuals with BD who were treated with valproate or lamotrigine was not related to the levels of any TRP catabolite.

Of the 17 participants taking lithium, there was only one responder, according to the established cutoff of 7 of the Alda scale. None of the 17 patients treated with valproic acid showed an adequate response. The mean of the eight individuals taking lamotrigine was a little higher but was still under the response cutoff. According to this result, a combination treatment with lithium and lamotrigine achieved the highest Alda scores for lithium as well as lamotrigine treatment, but this subsample was limited in size.

No differences were found in the TRP catabolites in the participants who showed a response to any mood-stabilizing medication versus the absence of response.

## 4. Discussion

This exploratory study demonstrated an association of a poorer response to lithium, according to the Alda scale, with higher circulating levels of KYN, KYN/TRP, as well as QA, which likely indicates the higher inflammatory and neurotoxic processes that occurred in those individuals with BD who had a poorer response to lithium. In contrast, the treatment response to the mood stabilizers valproate and lamotrigine was not related to TRP metabolites.

The finding that TRP is decreased in BD compared to HC is in line with the results of the meta-analyses by Marx et al. [25], as well as by Bartoli et al. [50]. An increased KYN/TRP reflecting the IDO-1 activity [51] in BD compared to HC has been shown by some studies [24,26,52,53], while other studies have not found a difference between the groups [54,55]. KYN and KYNA were found to be elevated in cerebrospinal fluid [56] and postmortem analyses [57]. Furthermore, higher 3-HK/KYNA levels, which reflect the KMO activity, were found in BD, which indicates a shift towards the neurotoxic branch [58]. Interestingly, QA, which is assumed to have neurotoxic features, was found to be lower in BD patients than in the HC in this study. Theoretically, the pro-inflammatory state triggers IDO-1 [17] and KMO [21], and thereby the conversion of TRP to KYN, and further, to 3-HK and QA. Furthermore, the accumulation of QA might reduce the KAT activity, which leads to less KYNA and, finally, to higher KMO and more QA [13]. Therefore, we expected to find higher KYN/TRP, 3-HK, and QA.

Unlike our initial hypothesis, the finding that individuals with BD showed no alteration in KYN or KYNA but had lower QA than the HC might be explained by the presence of long-term treatment with mood stabilizers and other drugs, which probably influences the biological processes, and thus the QA levels. Although the inflammatory alterations in BD were shown to be evident in depression, mania, and euthymia [16], there might be differences in the pro- and anti-inflammatory cytokines between these states [59], with effects on the TRP metabolism [60]. Besides the acute symptomatology, other variables, such as the course of illness (with number, entity [56], severity, and duration of former episodes), as well as psychosis [61] and suicidality [6,62,63], might also be of significant relevance for the inflammatory state and the TRP metabolism, and they should therefore be included in further studies. Additionally, the IDO-1 activity was shown to be increased in overweight and obesity [10,24], and was presumably indirectly driven via linked proinflammatory cascades [64]. The relatively low BMI of this sample might have also influenced the unexpected lack of differences. 

According to the literature, approximately 30% of individuals with BD are full responders to lithium [65]. However, we found a low response rate according to the dichotomous classification of the Alda scale in this cohort. Pharmacological combinations are frequently used, but the evidence is lacking, and it only gives a hint that lithium plus valproate might be as effective as monotherapy [66]. However, the measurement of the treatment response is generally based on a retrospective assessment. The measurement is therefore complicated by the subjectivity of the rater and the recall bias of the patient, and by the irregular course of the illness and the adherence [67]. Although lithium has been used for decades, there is still only a partial understanding of its mechanism of action. Studies in the 1980s suggested that the beneficial effect of lithium is 5-HT mediated [68], which refers to the increased levels of the 5-HT metabolite 5-hydroxyindoleacetic acid [69], and a higher risk for 5-HT syndrome under a combination therapy of lithium and 5-HT antidepressants [70]. Later, lithium was reported to have anti-inflammatory properties. An inhibiting effect on pro-inflammatory cytokines and lipopolysaccharides was found in microglial cells treated with lithium [71]. Because of the inhibition of glycogen synthase kinase 3 (GSK-3), not only the downregulation of pro-inflammatory genes, but also the upregulation of anti-inflammatory ones was shown in microglial cells [72].

This leads us to the assumption that lithium affects the TRP catabolism, presumably depending on the inflammatory processes. However, only a few in vitro studies have explored this issue. In the d-amphetamine rat model of mania, IDO-1 knockout and lithium or valproate treatments led to a decrease in manic symptoms that was associated with an attenuate effect of the mitochondrial stress [73]. Göttert et al. found an inhibiting effect of lithium on IDO-1 expression, as well as activity that was regulated by the inhibition of GSK-3 in immortalized human microglia [32]. These findings have to be interpreted with caution, as central processes in vivo might differ slightly because of the slow spread of lithium [74]. Furthermore, MRI studies show different lithium intensities in the brain regions [75]. Nevertheless, the results of this microglia study constitute a basis for the biological underpinnings for the results of our clinical study.

The lack of differences in the KYN/TRP ratio between individuals treated with lithium or other mood stabilizers or their combinations might be explained by the small sample size per group that was not calculated for this outcome. Our data suggest that the better the response to lithium according to the Alda score, the lower the conversion of TRP into KYN. Therefore, considering the inflammation-associated IDO-1 activity, these findings support the possible anti-inflammatory properties of lithium. Furthermore, the lower degree of the response to lithium was associated with higher QA levels in this study. Given that QA has neurotoxic features via the glutamatergic pathway, these findings strengthen the assumption that the clinical response might be associated with biochemical processes. In addition, the potential association of the clinical response to lithium with biochemical measurements could bring advantages in the selection of mood-stabilizing treatments. In the future, the modulation that is exerted by genetic factors [39], as well as the potential confounding effects that are determined by the phenotypic manifestations of the disorder (BD I versus BD II), will need to be further investigated.

Preclinical and clinical human studies show an influence of lithium on the MLT levels in several brain regions [33,76]. Underlying theories are the associations of the lithium to clock genes, as well as to the retinal–hypothalamic–pineal pathway that is involved in MLT secretion [28,33]. Consequently, lithium is highly relevant for the regulation of chronobiological rhythms, such as sleep–wake, the body temperature, the hormonal rhythms, and the appetite, whereas alterations are frequently evident in BD [77]. This can be partly integrated by our findings. The Alda scale only measures the effectiveness in episode prevention, and it is not focused on symptoms that pertain to biological rhythms. Therefore, one could conclude that the response to lithium is not dependent on the melatonergic functioning. Nonetheless, there might be differences between mood stabilizers in the MLT levels that we could not be shown because of some degree of heterogeneity in our sample.

Valproate was shown to impact the serotonergic neurotransmission, and in particular during mania [78]. Additionally, the degradation products of TRP towards the KYN pathway were investigated by a few animal studies. One study on rats shows that the anti-manic effect via the IDO-1 inhibition of lithium and valproate was the same [73]. Another trial found increases in TRP, KYN, and KYNA after valproate administration; however, these were mostly in central concentrations compared to plasma ones [79]. Lamotrigine was only investigated by one animal study, which showed an increasing effect on the KAT activity and, therefore, increased KYNA levels [80]. From a clinical perspective, our results did not show an impact of the response of the mood-stabilizing therapies with valproate or lamotrigine on the TRP catabolites.

However, the exploratory nature of the study paired with the small sample size do not allow firm conclusions. Future ad hoc studies that take advantage of the effect sizes that were found in our study are thus needed to finally address the link between the response to the different mood stabilizers and its possible correlation with the functioning of the KYN pathway.

### Limitations

There are several limitations of this study. First, the metabolites were measured in plasma. Although studies have concluded that there is a comparability to the central processes, there might be different levels because of the competing amino acids at the blood–brain barrier, especially for QA and KYNA [81]. Second, because of the naturalistic setting, the participants with BD were treated with different monotherapies, or with a combination of mood stabilizers at various durations. Therefore, the sample sizes of the groups taking a monotherapy were rather small. Moreover, the sample size of the main finding (namely, the lithium intake) was small, with *n* = 8 for monotherapy, and *n* = 9 for combination therapy. The findings of the differences between these groups must be interpreted with caution. However, given that this is the first study on this topic in BD, these preliminary data present a hint for clinical decision making and for further study planning through the provision of the effect sizes. Third, other drugs, which were not included in the analyses, might have an impact on TRP and its catabolites. However, controlling for every substance and combination was not statically feasible given the relatively small sample size. Fourth, the enzyme activity was derived from ratios. Hence, the degree of saturation and other co-factors might influence the accumulation of substrates. Fifth, we did not determine the other possible metabolites of TRP along the KYN pathway, including xanthurenic and picolinic acids, which may also be implicated in the pathophysiology and psychopharmacology of BD. Sixth, the possible differences in the response to treatment and its relationship with the KYN pathway, according to the diagnosis of bipolar I and II, were not investigated.

## 5. Conclusions

The present exploratory study investigated the metabolism of TRP via KYN in BD, and it displays higher plasma levels of KYN, KYN/TRP as a proxy for IDO-1 activity, and QA that are associated with the lower response to lithium, according to the Alda score. These exploratory findings indicate that the metabolites of the TRP-to-KYN pathway are worthy of further investigation as possible biomarkers of the treatment response to lithium in individuals with BD.

## Figures and Tables

**Figure 1 jcm-11-02517-f001:**
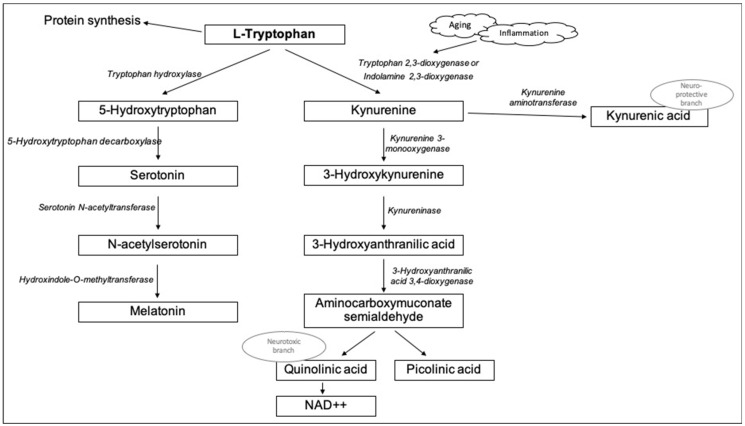
Biochemical pathway of tryptophan metabolism via serotonin and kynurenine.

**Figure 2 jcm-11-02517-f002:**
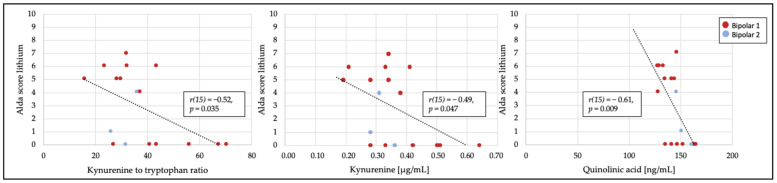
Significant correlations of the Alda score of lithium response to kynurenine/tryptophan ratio, kynurenine, and quinolinic acid.

**Table 1 jcm-11-02517-t001:** Sociodemographic and illness-specific data of study participants.

	Individuals with BD (*n* = 48)	HC (*n* = 48)	Statistics	*p*	Effect Size
Age [years]	51.59 (±11.13)	51.71 (±8.47)	*t* (87,14) = −0.06	0.951	*d* = 0.01
Sex	37.5% male 62.5% female	37.5% male 62.5% female	*X*^2^ = 0	1.00	
BMI [kg/m^2^]	25.33 (±5.24)	23.60 (±3.42)	*t* (94) = 1.91	0.059	*d* = 0.39
Smoking status	27.1% no 47.9% yes 25.0% ex-smoker	68.8% no 18.8% yes 12.5% ex-smoker	*X*^2^ = 16.82	**<0.001 ****	***η*^2^ = 0.419**
TRP [μg/mL]	9.54 (±1.64)	11.06 (±1.69)	*F* (1,91) = 18.26	**<0.001 ****	***η*^2^ = 0.167**
KYN [μg/mL]	0.35 (±0.10)	0.38 (±0.14)	*F* (1,91) = 2.070	0.154	*η*^2^ = 0.022
KYN/TRP*1000 ratio	38.11 (±14.50)	34.86 (±11.97)	*F* (1,91) = 0.99	0.321	*η*^2^ = 0.011
KYNA [ng/mL]	9.02 (±4.76)	10.25 (±8.09)	*F* (1,91) = 1.26	0.264	*η*^2^ = 0.014
3-HK [ng/mL]	42.46 (±2.84)	44.40 (±13.48)	*F* (1,91) = 0.86	0.356	*η*^2^ = 0.009
QA [ng/mL]	142.62 (±15.13)	158.73 (±45.89)	*F* (1,91) = 4.91	**0.029 ***	***η*^2^ = 0.051**
QA/KYNA ratio	24.01 (±20.93)	22.54 (±14.01)	*F* (1,91) = 0.24	0.623	*η*^2^ = 0.003
5-HTP [ng/mL]	87.90 (±29.49)	69.93 (±34.48)	*F* (1,91) = 5.95	**0.017 ***	***η*^2^ = 0.061**
5-HT [ng/mL]	319.25 (±168.83)	260.74 (±167.97)	*F* (1,91) = 3.55	0.063	*η*^2^ = 0.038
MLT [pg/mL]	11.44 (±6.53)	11.89 (±5.82)	*F* (1,91) = 0.09	0.764	*η*^2^ = 0.001

Note. Abbr.: BD = bipolar disorder; BMI = body mass index; HC = healthy controls; KYN = kynurenine; KYNA = kynurenic acid; QA = quinolinic acid; TRP = tryptophan; 3-HK = 3-hydroxykynurenine; 5-HTP = 5-hydroxytryptophan; 5-HT = serotonin; MLT = melatonin. Significant results are presented in bold letters; * *p* < 0.05, ** *p* < 0.01. Data are reported as means (±standard deviations).

**Table 2 jcm-11-02517-t002:** Clinical variables and mood-stabilizing treatment of participants with BD.

	Individuals with BD (*n* = 48)			
Diagnosis	Bipolar I: 72.9% Bipolar II: 27.1%			
Years of illness (Mean ± SD)	20.82 (±9.72)			
Number of depressions (Mean ± SD)	6.42 (±9.59)			
- Mild	4.71 (±4.66)			
- Moderate	4.87 (±8.93)			
- Severe	1.56 (±2.03)			
Number of manic episodes (Mean ± SD)	6.78 (±10.19)			
- Hypomanic	3.42 (±6.10)			
- Moderate manic	2.38 (±4.74)			
- Severe manic	1.13 (±1.85)			
CGI (Mean ± SD)	2.53 (±0.91)			
HAMD (Mean ± SD)	4.11 (±3.86)			
YMRS (Mean ± SD)	1.68 (±2.49)			
HAMA (Mean ± SD)	6.21 (±5.14)			
		**Alda Score**		
		Lithium	Valproate	Lamotrigine
Current prophylactic treatment (*n*, %)	44 (91.66%)			
- only lithium	8 (16.67%)	2.75 (±2.76)		
- only anticonvulsant	11 (22.92%)			
- only valproate	8 (16.67)%		4.00 (±0.82)	
- only lamotrigine	2 (4.17%)			1.00 (±1.41)
- only antipsychotic	3 (6.25%)			
- lithium + anticonvulsant	7 (14.58%)	3.86 (±2.79)		
- with valproate	3 (6.25%)	3.67 (±3.21)	3.67 (±3.21)	
- with lamotrigine	2 (4.17%)	5.50 (±2.12)		8 (±0)
- lithium + antipsychotic	2 (4.17%)	0		
- anticonvulsant + antipsychotic	10 (20.83%)			
- with valproate	7 (14.58%)		2.86 (±2.48)	
- with lamotrigine	1 (2.08%)			7
Other psychopharmaceuticals (*n,* %)				
- Antipsychotics in subprophylactic dose (*n* = 44)	17 (38.64%)			
- Antidepressants (*n* = 44)	11 (25.00%)			
- Benzodiazepines (*n* = 44)	23 (52.27%)			

Note. Abbr.: BD = bipolar disorder; CGI = Clinical Global Impression; HAMA = Hamilton Rating Scale for Anxiety; HAMD = Hamilton Rating Scale for Depression; SD = standard deviation; YMRS = Young Mania Rating Scale.

## Data Availability

The data presented in this study are available upon request from the corresponding author, because of ongoing analysis.

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
