# Peer review of "Is Poor Lithium Response in Individuals with Bipolar Disorder Associated with Increased Degradation of Tryptophan along the Kynurenine Pathway? Results of an Exploratory Study"

_jcm, 2022, doi:10.3390/jcm11092517_

Round 1

Reviewer 1 Report

The authors present the the study "Poor lithium response in individuals with bipolar disease is associated with increased degradation of tryptophan into the kynurenine pathway".

Aim of the study was (1) to compare tryptophan (TRP) catabolite levels in individuals with BD and healthy controls, (2) investigate associations with current psychiatric symptomatology and illness history and (3) response to lithium, valproate and lamotrigene.

These hypotheses are based on the assumption that the catabolism of TRP impacts "the genesis, symptoms and outcome of neuropsychiatric disorders such as BD" (line 67-68). As summed up in the introduction, TRP catabolism in BD is assumedly increased by an inflammation-mediated upregulation of IDO, leading to lower TRP and lower metabolites in the 5-HT pathway. As KAT is expected to be suppressed in inflammation, the increased KYN metabolism should lead to higher levels of QA.

The study analysed then 48 euthymic patients with BD and 48 healthy controls.

The results for hypothesis 1 showed lower TRP, higher 5-HTP and lower QA in BD. The results for hypothesis 2 were a correlation of KYN/TRP and number of depressive  and hypomanic episodes. For the latter, even an association with higher QA-levels was found. QA correlated negatively with years of illness. There was a correlation between 5-HT levels with the number of suicide attempts and number of depressive episodes with psychotic features. The results for hypothesis 3 were correlations of KYN, KYN/TRP and QA with treatment-response scores in various subgroups regarding treatment with lithium. Current treatment did not show differences in any catabolite.

The authors conclude, that higher plasma levels of KYN, KYN/TRP as a proxy for IDO-1 activity as well as QA is associated with lower response to lithium according to the treatment-response score.

General considerations:

The current work was intended to study changes in the TRP metabolism as a consequence of inflammation-mediated processes in BD. This included

a) differences between patients with BD and healthy controls

b) differences resulting from the current mood episode in BD

c) differences in historic mood episodes in patients with BD

d) differences according to current treatment.

The the data presented to a) do not support the postulated mechanism but its opposite. b) cannot be studied with the cohort of euthymic patients and needs to be taken out of the paper. The findings to c) are very difficult to interpret with the data presented, see below. Concerning d), the authors did not find any impact on TRP metabolites by treatment, but a correlation with a treatment response score. Of note, only a single patient hat a good response under lithium treatment. To generalize the findings in the current form overstresses the data quality.

Specific considerations

Abstract:
It is unclear where the number 50 is coming from. 

Introduction:
- a figure illustrating the pathways would be very helpful in understanding the reasoning.

Method:
- Selecting criteria/recruitment of the BD cohort needs to be described.
- Any power analysis available to motivate the number 48?
- The HC are apparently matched (and very accurately so). Describe matching and recruitment
- specify how assessment was done: medical records or patient interview
- describe procedure of rating the Alda scale: Who rated? Single person? At what time? Two-year-period pre-study?

Results:
Table 2: Anticonvulsants seem to have ended in wrong row.  Percentages do not sum up: other anticonvulsants than Val and Lam? How is it possible for patients with a combination of Li and Lam to have different Alda - scores for Li and Lam?
Any data on life time-lithium treatment duration available?

Author Response

Q1) General considerations:

The current work was intended to study changes in the TRP metabolism as a consequence of inflammation-mediated processes in BD. This included

  1. a) differences between patients with BD and healthy controls

  1. b) differences resulting from the current mood episode in BD

  1. c) differences in historic mood episodes in patients with BD

  1. d) differences according to current treatment.

The the data presented to a) do not support the postulated mechanism but its opposite. b) cannot be studied with the cohort of euthymic patients and needs to be taken out of the paper. The findings to c) are very difficult to interpret with the data presented, see below. Concerning d), the authors did not find any impact on TRP metabolites by treatment, but a correlation with a treatment response score. Of note, only a single patient hat a good response under lithium treatment. To generalize the findings in the current form overstresses the data quality.

  1. R) Thank you for your time in assessing our manuscript and your valuable input. Concerning point a) you are correct that we expected to find an elevated degradation of TRP towards KYN and its catabolites only in BD compared to HC, but even if these results are partly conflicting we think they are worth to publish as the literature remains incomplete and needs further insights. As mentioned in the discussion we consider that the pharmacological treatment, the relatively low BMI and the euthymic state might have influenced these results. Concerning point b) we agree with you that the interpretation of the current symptomatology is insufficient in this euthymic cohort and therefore deleted in the results and discussion sections. Regarding c) we have revised and streamlined the paper in accordance with the reviewers’ observation to improve readability and interpretability. Point d) is also well taken but we would like to observe that the scale score was used as a continuous trait, which in our population of patients ranged from 0 to 7. The presence of only one patient with a response score equal or higher than 7 (good responder) does not limit the identification of an association between an independent variable and a higher degree of response. Clearly, these are exploratory analyses that await confirmation in independent and larger studies.

Q2) Abstract:It is unclear where the number 50 is coming from.

R2) We apologize for this mistake derived from former analyses including not only the euthymic patients. This has now been corrected to n=48.

Q3) Introduction: a figure illustrating the pathways would be very helpful in understanding the reasoning.

R3) According to your suggestions we added a figure (Figure 1) presenting the tryptophan metabolism in the introduction.

Q4) Selecting criteria/recruitment of the BD cohort needs to be described.

  1. R) More details were added in the text.

Q5) Any power analysis available to motivate the number 48?

R5) As reported previously in our study protocol,1  a sample size of 50 individuals with BD and 50 HC was considered adequately powered to detect a difference between BD and HC in melatonin levels and related compounds to our primary goal of studying the metabolism of TRP along the 5-HT/melatonin and KYN pathways in BD. For the sake of this analysis we included 48 patients for whom we had complete clinical and laboratory measures. This issue has been clarified in the methods: “This sample size was deemed adequately powered (more than 95%) to detect a difference between groups, with an α set at 0.05 considering an effect size in the difference of MLT levels between HC and BD patients equal to 5.3 (see [38] for further details).”

Q6) The HC are apparently matched (and very accurately so). Describe matching and recruitment.

R6) This has been clarified in the text: “HC were recruited by word of mouth among hospital staff, their families, and university students. They underwent a standard medical and laboratory test assessment to verify their health status and clinically assessed for establishing the absence of any previous and current mental disorder. Furthermore, cases and controls were matched for age and sex”.

Q7) Specify how assessment was done: medical records or patient interview.

R7) This has been clarified in the text: “Diagnoses and clinical assessments were performed by trained psychiatrists through direct interview and a systematic review of patients medical records.”

Q8) Describe procedure of rating the Alda scale: Who rated? Single person? At what time? Two-year-period pre-study?

R8) The assessment of clinical response to mood stabilizers was performed by trained psychiatrists with the supervision of one senior rater (M.M.) who has worked in the validation procedure of the scale.We performed a detailed revision of the charts to permit graphic depiction of the longitudinal clinical course with the lifechart method. In our study this included both the retrospective assessment of past clinical course (based on accurate longitudinally collected clinical data) and the prospective two-year observation. Then we calculated the area under the curve of illness activity (severity of the episodes x duration of episodes) before and after the introduction of a mood stabiliser or combination of mood stabilisers. This gave us an objective measure of clinical improvement under a specific treatment or combination of treatments (criterion A of the scale). This was added to the methods: “Specifically, we performed a detailed revision of the clinical charts to permit graphic depiction of the longitudinal clinical course with the life chart method. In our study this included both the retrospective assessment of past clinical course (based on accurate longitudinally collected clinical data) and the prospective two-year observation. Then we calculated the area under the curve of illness activity (severity of the episodes x duration of episodes) before and after the introduction of a mood stabiliser or combination of mood stabilisers. This gave us an objective measure of clinical improvement under a specific treatment or combination of treatments (criterion A of the scale). The assessment of clinical response to mood stabilizers was performed by trained psychiatrists with the supervision of one senior rater (M.M.) who has worked in the validation procedure of the scale [41].”

Q9) Table 2: Anticonvulsants seem to have ended in wrong row. Percentages do not sum up: other anticonvulsants than Val and Lam?

  1. R) There were also three patients taking carbamazepine and one topiramate and one oxcarbazepine, and this has been clarified in the text. Response ratings according to the Alda scale was however only conducted for lithium, valproate, and lamotrigine.

Q10) How is it possible for patients with a combination of Li and Lam to have different Alda - scores for Li and Lam?

R10) The Alda scale is validated for the assessment of response to mood stabilizers (lithium, anticonvulsants, and atypical antipsychotics). In this naturalistic study some patients took combinations of mood stabilisers for variable duration, or were treated sequentially with diverse mood stabilizers. In this case we are expressing the improvement observed under each mood stabiliser and not concomitant treatment. This was clarified in the methods: “In this naturalistic study some patients took combinations of mood stabilisers for variable duration, or were treated sequentially with diverse mood stabilizers. Here we are expressing the improvement observed under each mood stabiliser and not concomitant treatment.”

Q11) Any data on life time-lithium treatment duration available?

R11) Length of lithium treatment was on average 10 years with a standard deviation of +/- 9.2 years. This was added to the main text: “Patients had been treated with lithium for an average of 10 years with a standard deviation of +/- 9.2 years.”

Reviewer 2 Report

The current paper tries to address interesting research questions. However, these are clearly post-hoc analyses on an inappropriate dataset to investigate the research questions raised. The title is plainly misguiding, the methodology section is very incomplete, statistics are suboptimal, and the results section needs to be more structured and complete in reporting of the performed analyses. 

Title:

The title is misguiding. It represents a posthoc analysis result, and does not reflect the main goal of the study.

Introduction:

The authors present a simplified version of the pathway. Nowadays, other metabolites such as xanthurenic acid, picolinic acid and quinaldic acid are typically included.

First generation antipsychotics, especially haloperidol, are also first-line choices in the treatment of bipolar disorder

The focus of the research questions is not so clear. Does it aim to investigate kynurenine and serotonine pathways? Does it want to investigate the impact of symptom severity, duration of illness, number of mood episodes in the last two year? Or is the main outcome the impact of treatment response, impact of lithium versus valproate versus lamotrigine? Or all of the above?

Methods:

  • Was the Alda score based on an interview by the researchers, thus basing the score on treatment response on reporting of the patient only? Was any information from clinicians used? Were there any objective assessment of treatment compliance? It seems a very biased method to assess adequate response on medication.
  • Did the authors perform a power analysis? They investigate the impact of medication, but 16 lithium users, 16 valproate users and 8 lamotrigine users are sample sizes that seem too low for the current research questions.

Laboratory methods & data preparation:

  • describe which lab did the analyses
  • how were samples stored, What was the time interval between storage and analyses?
  • Were there freeze-thaw cycles?
  • describe LCMS/MS analyses more in depth
  • were analyses done in duplo/triplo? What were CV’s of the analytes? Which CV-cut off did the authors apply?
  • How much percentage of each of the analytes were above detection limit?
  • How did authors deal with outliers?

Statistical analyses

  • did the authors control for smoking status? That has repetitively been shown to be a covariate, sometimes the main variable driving inflammatory abnormalities between patients and controls
  • The number of different combinations of  investigated drugs is high (see table 2). Apart from the fact that this simply is not an adequate dataset to investigate the impact of medication, MANCOVA analyses are not indicated for such data.
  • Lithium is taken in monotherapy by only 16% (n=8?) of the patients, same for valproate (n=7-8?) and lamotrigine by only 4(!)% (n=2?). The current study design and patient sample is simply not adequate to address these research questions.
  • The use of mood-stabilising antipsychotics (taken by 30% of the sample; apart from antipsychotics taken in sub-prophylactic dosages) is ignored in the analyses. How do the authors know the efficacy of the treatment is due to lithium/antiepileptics and not due to antipsychotics like olanzapine/quetiapine/haloperidol/... given that these are also first line choice medications in BD patients?

Results

In general the results section is a bit chaotic and seems like a highly selective reporting of the positive findings, but fails to report on performed analyses that were negative. The analyses performed to investigate the impact of lithium, valproate and lamotrigine are very unclear, and hardly reported. The results sections could use more structuring according to the several research questions.

  • Table 1: are numbers between brackets SD’s or SE’s
  • Correlations between number of depressive episodes and KYN/TRP are reported. However, please provide a correlational table indicating which correlations were investigated. If 10 metabolites are correlated with number depressions/manic episodes (+ all subcategories), depression/mania ratio, psychotic episodes and suicide attempts, we are talking about 90 correlation coefficients that have been explored, so only reporting the 6-7 borderline significant associations out of 90 performed correlational analyses gives a high risk for Type I errors, so full reporting is crucial.
  • Table 2: do I understand it correctly that while the impact of lithium on tryptophan catabolites seems to be the main focus of the paper (cfr title), only 33% of the patients included in the study actually took lithium (i.e. 15-16 patients)? What is the main focus of this paper?
  • Please provide numbers on top of percentage of patients taking the investigated medication
  • Table 2 is very confusing. It should clearly be stated in the results section how many patients took lithium, lamotrigine and valproate. Given that these products seem to be the focus of the paper (cfr last paragraph of the introduction), exclude patients not taking these drugs (i.e. 15%?).
  • As already indicated, it is hard to investigate the impact of these drugs while there are several of these pharmaceuticals being given in all kinds of combinations. This is just not the right design to investigate the questions raised by the authors. The sample is underpowered
  • Moreover, regression analyses seem more suitable to approach these data, even when not considering the lack of statistical power
  • Were blood levels determined for valproate/lithium/lamotrigine?
  • Results of the mancova analyses investigating medication effects are not actually presented.
  • Figure 1: what are r and p values per figure?

Discussion

  • The results being put forward as the main results are very skewed and selective, as they represent the results of one of the tertiary outcome questions, with 15 or 16 patients out of the 48 patients entering these analyses.
  • The discussion is unfocused, overinterpretative and lacks critical appraisal of their own limitations. It is not enough to say that results have to be interpreted with caution.

Author Response

Q1) The current paper tries to address interesting research questions. However, these are clearly post-hoc analyses on an inappropriate dataset to investigate the research questions raised. The title is plainly misguiding, the methodology section is very incomplete, statistics are suboptimal, and the results section needs to be more structured and complete in reporting of the performed analyses.

  1. R) Thanks for your time and detailed remarks. We have revised the manuscript in a point by point fashion. Further, we have focused in a more structured way the main aim of the study, namely the association between the response to mood-stabilizing treatment and tryptophan catabolites. We have also added a figure to illustrate the metabolism of TRP, expanded the methods section, removed the analyses with current and former BD symptomatology from the results and discussion, and shortened the discussion as well as added limitations.

 Q2) The title is misguiding. It represents a posthoc analysis result, and does not reflect the main goal of the study.

R2) As described above, our main aim was to analyse the association between response to mood-stabilizing treatment and tryptophan catabolites. We concur with the invitation to moderate our claims and the title has been revised as follows: “Is poor lithium response in individuals with bipolar disorder associated with increased degradation of tryptophan along the kynurenine pathway? Results from an exploratory study”

Q3) The authors present a simplified version of the pathway. Nowadays, other metabolites such as xanthurenic acid, picolinic acid and quinaldic acid are typically included.

R3) We thank the reviewer for this comment. It is true that many other metabolites are formed along the kynurenine pathway, which is now illustrated in a figure in the introduction. According to current knowledge and several works done by our group on the tryptophan metabolism in mood disorders and comorbidities, we have determined those metabolites that seem to be associated with the pathophysiology of bipolar disorder. However, future studies are warranted to examine the possible contribution of other metabolites of the metabolism of Tryptophan via kynurenine in the pathophysiology and psychopharmacology of bipolar disorder. This lack of studies has been now mentioned in the limitations “Fifth, we did not determine other possible metabolites of TRP along the KYN pathway including xanthurenic and picolinic acids that may also be implicated in the pathophysiology and psychopharmacology of BD”.

Q4) First generation antipsychotics, especially haloperidol, are also first-line choices in the treatment of bipolar disorder

R4) They are indeed, but only for acute episodes and certainly not for long term maintenance treatment, which is the focus of our work. Please see reference 3.

Q5) The focus of the research questions is not so clear. Does it aim to investigate kynurenine and serotonine pathways? Does it want to investigate the impact of symptom severity, duration of illness, number of mood episodes in the last two year? Or is the main outcome the impact of treatment response, impact of lithium versus valproate versus lamotrigine? Or all of the above?

R6) We thank you for this comment. To clarify the main focus we deleted the findings and discussions of the associations with the current as well as former illness symptomatology and focus only on first the comparison of individuals with BD versus HC and second on the associations with response to lithium, valproate and lamotrigine. We have made clearer the context of our study and the scientific goal at the end of the Introduction.

Q7) Was the Alda score based on an interview by the researchers, thus basing the score on treatment response on reporting of the patient only? Was any information from clinicians used? Were there any objective assessment of treatment compliance? It seems a very biased method to assess adequate response on medication.

R7) The Alda scale is the only validated method to assess response to long term treatment with mood stabilisers. It has limitations but if applied to accurate longitudinal clinical information has adequate reliability.2 We concur with the reviewer that more detail are needed and we have added an extensive explanation: “Specifically, we performed a detailed revision of the clinical charts to permit graphic depiction of the longitudinal clinical course with the life chart method. In our study this included both the retrospective assessment of past clinical course (based on accurate longitudinally collected clinical data) and the prospective two-year observation. Then we calculated the area under the curve of illness activity (severity of the episodes x duration of episodes) before and after the introduction of a mood stabiliser or combination of mood stabilisers. This gave us an objective measure of clinical improvement under a specific treatment or combination of treatments (criterion A of the scale). The assessment of clinical response to mood stabilizers was performed by trained psychiatrists with the supervision of one senior rater (M.M.) who has worked in the validation procedure of the scale [41].”

Q8) Did the authors perform a power analysis? They investigate the impact of medication, but 16 lithium users, 16 valproate users and 8 lamotrigine users are sample sizes that seem too low for the current research questions.

R8) As highlighted in the answer to comment #5 by reviewer #1, the sample size was based on a previous power analysis (Manchia et al., 2019;7:27). Since the main goal of that study was not to look at the relationship of peripheral biomarkers with treatment response, the study could be underpowered. This limitation has been clearly mentioned in the manuscript. However, given that this is the first study on this topic in BD, this data will be essential for calculating the sample size in future studies designed for such outcome. We have clarified this in the text: “This sample size was deemed adequately powered (more than 95%) to detect a difference between groups, with an α set at 0.05 considering an effect size in the difference of MLT levels between HC and BD patients equal to 5.3 (see [38] for further details).”

Q9) describe which lab did the analyses

R9) The analysis of TRP metabolites were conducted in the laboratory of Prof. Stefano Comai at the University of Padova according to standard procedures. The lab of the University of Padova now headed by S. Comai has a long-lasting and internationally recognised expertise in the study of the metabolism of TRP via 5-HT and KYN in pathophysiological conditions (The lab was founded almost 60 years ago by late Prof. Luigi Musajo, and then run by Prof. Graziella Allegri until 12 years ago, two pioneers in Tryptophan research). Few references to our recent works in which we determined the metabolites on TRP via 5-HT and KYN have been reported.

Q10) how were samples stored, What was the time interval between storage and analyses?

R10) Thank you for this comment. Indeed, we did not properly report the methods. We have now included this information: “Plasma was aliquoted and stored at -80 °C until the analysis.”

Q11) Were there freeze-thaw cycles?

R11) Plasma samples were thawed just prior to the HPLC analyses.

Q12) describe LCMS/MS analyses more in depth

R12) As mentioned previously, the analysis of TRP metabolites via 5-HT and KYN is a routine in our lab and we have reported the methodology in many papers during the last decade/s. We slightly revised this section, but we think this information can be easily found by the readers in our recent work described in references number [20,48,49]. Importantly, some of these are Open Access, so accessible to everyone interested (e.g. see ref #48 Tryptophan Metabolites, Cytokines, and Fatty Acid Binding Protein 2 in Myalgic Encephalomyelitis/Chronic Fatigue Syndrome. Biomedicines 2021, 9, 1724, doi:10.3390/biomedicines9111724). However, if the editor and/or the reviewer think that we should provide in any case all the methodological details, we will be very happy to add this information.

Q13) were analyses done in duplo/triplo? What were CV’s of the analytes? Which CV-cut off did the authors apply?

R13) According to our original method (Rapid determination of tryptophan and its metabolites along the kynurenine pathway by HPLC, in: Progress in Tryptophan and Serotonin Research, W. de Gruyter, Berlin, 1984, 67-70, C.Costa, A.Bettero, G.Allegri) which over the years has only undergone little modifications that were included in the references mentioned above in Q12, analyses were done singularly.

Q14) How much percentage of each of the analytes were above detection limit?

R14) No analytes were below the detection limit. With this method we were able to quantify all the metabolites investigated in this study.

Q15) How did authors deal with outliers?

R15) We did not have outliers for both biological and psychometric variables. Normality was tested with the Kolmogorov-Smirnov test prior computing parametric statistics described in the Methods. This info was missing and has been now included. Thank you for highlighting this issue.

Q16) did the authors control for smoking status? That has repetitively been shown to be a covariate, sometimes the main variable driving inflammatory abnormalities between patients and controls

R16) We agree with that point. We referenced to the potential cofounding factors in the introduction: “The tryptophan degradation is affected by sex [7], age [8], smoking [9], weight [10] and somatic as well as mental health status.” Additionally the analyses were controlled for smoking status (currently yes/no, ex-smoker) but however were not affected by this.

Q17) The number of different combinations of investigated drugs is high (see table 2). Apart from the fact that this simply is not an adequate dataset to investigate the impact of medication, MANCOVA analyses are not indicated for such data.

R17) Indeed the interpretation of the possible interactions and cofounding effects with these combination treatments should be conducted with caution. Due to the high number of mono-/ combination treatment groups the second MANCOVA was therefore excluded from the manuscript and the focus set on the correlation with the treatment response.

Q18) Lithium is taken in monotherapy by only 16% (n=8?) of the patients, same for valproate (n=7-8?) and lamotrigine by only 4(!)% (n=2?). The current study design and patient sample is simply not adequate to address these research questions.

R18) The sample size is small and the study exploratory in nature. Thus we have added this limitation in the Discussion: “Due to the naturalistic setting, participants with BD were treated with different monotherapy or combination of mood-stabilizers to various durations. Therefore, the sample sizes of the groups taking a monotherapy were rather small. Moreover, the sample size of the main finding namely lithium intake was small with n=8 for monotherapy and n=9 for combination therapy. The findings of differences between these groups have to be interpreted with caution”.

Q19) The use of mood-stabilising antipsychotics (taken by 30% of the sample; apart from antipsychotics taken in sub-prophylactic dosages) is ignored in the analyses. How do the authors know the efficacy of the treatment is due to lithium/antiepileptics and not due to antipsychotics like olanzapine/quetiapine/haloperidol/... given that these are also first line choice medications in BD patients?

R19) This is now clearly explained in the methods: “Specifically, we performed a detailed revision of the clinical charts to permit graphic depiction of the longitudinal clinical course with the life chart method. In our study this included both the retrospective assessment of past clinical course (based on accurate longitudinally collected clinical data) and the prospective two-year observation. Then we calculated the area under the curve of illness activity (severity of the episodes x duration of episodes) before and after the introduction of a mood stabiliser or combination of mood stabilisers. This gave us an objective measure of clinical improvement under a specific treatment or combination of treatments (criterion A of the scale). The assessment of clinical response to mood stabilizers was performed by trained psychiatrists with the supervision of one senior rater (M.M.) who has worked in the validation procedure of the scale [41].”

Q20) In general the results section is a bit chaotic and seems like a highly selective reporting of the positive findings, but fails to report on performed analyses that were negative. The analyses performed to investigate the impact of lithium, valproate and lamotrigine are very unclear, and hardly reported. The results sections could use more structuring according to the several research questions.

R20) We hope the deletion of the findings of the clinical symptomatology (also in table 2) led to an easier reading of the results as well as discussion section. The main analyses are the correlations of the measures of TRP catabolites with response to lithium, valproate and lamotrigine.

Q21) Table 1: are numbers between brackets SD’s or SE’s

R21) We apologize for this potential misunderstanding and in the note of the table “ Data are reported as mean  (±Standard Deviation).”

Q22) Correlations between number of depressive episodes and KYN/TRP are reported. However, please provide a correlational table indicating which correlations were investigated. If 10 metabolites are correlated with number depressions/manic episodes (+ all subcategories), depression/mania ratio, psychotic episodes and suicide attempts, we are talking about 90 correlation coefficients that have been explored, so only reporting the 6-7 borderline significant associations out of 90 performed correlational analyses gives a high risk for Type I errors, so full reporting is crucial.

R22) With the deletion of the historic illness episodes and symptomatology this has been solved.

Q23) Table 2: do I understand it correctly that while the impact of lithium on tryptophan catabolites seems to be the main focus of the paper (cfr title), only 33% of the patients included in the study actually took lithium (i.e. 15-16 patients)? What is the main focus of this paper?

R23) The main aim of the paper was to analyse the association of tryptophan catabolites and the treatment response of the three mainly used mood-stabilizing medication lithium, valproate and lamotrigine. As only lithium response was found to be relevant for the tryptophan towards the kynurenine and further quinolinic acid degradation these results were addressed in the title.

Q24) Please provide numbers on top of percentage of patients taking the investigated medication

R24) Please find the added numbers in the table.

Q25) Table 2 is very confusing. It should clearly be stated in the results section how many patients took lithium, lamotrigine and valproate. Given that these products seem to be the focus of the paper (cfr last paragraph of the introduction), exclude patients not taking these drugs (i.e. 15%?).

R25) Hopefully the added numbers can clarify the samples in an appropriate manner. Additionally, the numbers and reasons of exclusion of patients taking no or other medication than lithium, valproate and/or lamotrigine are described in the text. 

Q26) As already indicated, it is hard to investigate the impact of these drugs while there are several of these pharmaceuticals being given in all kinds of combinations. This is just not the right design to investigate the questions raised by the authors. The sample is underpowered

R26) We agree with the reviewer that it seems difficult to interpret these naturalistic data and that a larger cohort with a homogeneous monotherapy would be desirable, but we feel confident that the presentation and discussion of data of clinical studies and exploratory data present a benefit for further study planning and is therefore worth to publish. In the present study, we investigated for the first time that response to lithium might be involved in tryptophan catabolism processes. However, as you suggested also in other comments, we moderated the interpretation and discussed the results with more critical appraisal. Additionally, we added some limitations.

Q27) Moreover, regression analyses seem more suitable to approach these data, even when not considering the lack of statistical power

R27) We agree with the reviewer that we previously did not clarify properly the aim of our study and thus a multiple regression analysis approach could theoretically be more suitable given that we studied multiple independent variables over multiple dependent variables. We have now simplified the manuscript focussing on the main objective, e.g. the possible relationship between treatment response and the KYN pathway. For this objective, we believe that the correlation analyses could give us a better estimation of the possible association between TRP catabolites and treatment response without overinterpreting our findings.

Q28) Were blood levels determined for valproate/lithium/lamotrigine?

R28) Monitoring of blood levels is performed as standard procedures and these data are included in the assessment of response to mood stabilisers, and specifically item B4 of the scale.

Q29) Results of the mancova analyses investigating medication effects are not actually presented.

R29) Please see response to Q17.

Q30) Figure 1: what are r and p values per figure?

R30) The statistical values presented in the text are referred to the figure. According to your remark, we also added them to the figure.

Q31) The results being put forward as the main results are very skewed and selective, as they represent the results of one of the tertiary outcome questions, with 15 or 16 patients out of the 48 patients entering these analyses.

R31) This limitation is now mentioned in the manuscript.

Q32) The discussion is unfocused, overinterpretative and lacks critical appraisal of their own limitations. It is not enough to say that results have to be interpreted with caution.

R32) We have revised and streamlined the Discussion.

References

1) Manchia M, Squassina A, Pisanu C, Congiu D, Garzilli M, Guiso B, Suprani F, Paribello P, Pulcinelli V, Iaselli MN, Pinna F, Valtorta F, Carpiniello B, Comai S. Investigating the relationship between melatonin levels, melatonin system, microbiota composition and bipolar disorder psychopathology across the different phases of the disease. Int J Bipolar Disord. 2019;7(1):27.

2) Manchia M, Adli M, Akula N, Ardau R, Aubry JM, Backlund L, Banzato CE, Baune BT, Bellivier F, Bengesser S, Biernacka JM, Brichant-Petitjean C, Bui E, Calkin CV, Cheng AT, Chillotti C, Cichon S, Clark S, Czerski PM, Dantas C, Zompo MD, Depaulo JR, Detera-Wadleigh SD, Etain B, Falkai P, Frisén L, Frye MA, Fullerton J, Gard S, Garnham J, Goes FS, Grof P, Gruber O, Hashimoto R, Hauser J, Heilbronner U, Hoban R, Hou L, Jamain S, Kahn JP, Kassem L, Kato T, Kelsoe JR, Kittel-Schneider S, Kliwicki S, Kuo PH, Kusumi I, Laje G, Lavebratt C, Leboyer M, Leckband SG, López Jaramillo CA, Maj M, Malafosse A, Martinsson L, Masui T, Mitchell PB, Mondimore F, Monteleone P, Nallet A, Neuner M, Novák T, O'Donovan C, Osby U, Ozaki N, Perlis RH, Pfennig A, Potash JB, Reich-Erkelenz D, Reif A, Reininghaus E, Richardson S, Rouleau GA, Rybakowski JK, Schalling M, Schofield PR, Schubert OK, Schweizer B, Seemüller F, Grigoroiu-Serbanescu M, Severino G, Seymour LR, Slaney C, Smoller JW, Squassina A, Stamm T, Steele J, Stopkova P, Tighe SK, Tortorella A, Turecki G, Wray NR, Wright A, Zandi PP, Zilles D, Bauer M, Rietschel M, McMahon FJ, Schulze TG, Alda M. Assessment of Response to Lithium Maintenance Treatment in Bipolar Disorder: A Consortium on Lithium Genetics (ConLiGen) Report. PLoS One. 2013 ;8(6):e65636.

3) Yatham LN, Kennedy SH, Parikh SV, Schaffer A, Bond DJ, Frey BN, Sharma V, Goldstein BI, Rej S, Beaulieu S, Alda M, MacQueen G, Milev RV, Ravindran A, O'Donovan C, McIntosh D, Lam RW, Vazquez G, Kapczinski F, McIntyre RS, Kozicky J, Kanba S, Lafer B, Suppes T, Calabrese JR, Vieta E, Malhi G, Post RM, Berk M.

Canadian Network for Mood and Anxiety Treatments (CANMAT) and International Society for Bipolar Disorders (ISBD) 2018 guidelines for the management of patients with bipolar disorder. Bipolar Disord. 2018;20(2):97-170.

Reviewer 3 Report

This is a clinically interesting study, relevant to pharmacologists, and drug developers, but also to clinicians

However, there are some queries that require addressing.

1) It remains unclear how the sample was recruited, for instance randomly or consecutively .

2) There are 48/50 individuals in each group. Why were 4 individuals, 2 in each group excluded? How were they statistically handled?

3) Expand on sample size calculation: 5.3 refers to the F value of 5.29 from Kennedy 1996 which was then used by Manchia et al 2019 to investigate differences in melatonin concentrations. The current study however examines KYN/TRP, KYN and QA. How were effect sizes defined?

4) Please explain the Alda scale more clearly, it may be worth to include a text box with an example. Please explain in how far the Alda scale that was designed to measure the treatment effect of lithium can be used for other mood stabilisers or drug combos.

5) In this study, lithium augmented by lamotrigine was more effective than lithium on its own. Please discuss the reasons for that. For figure 2 you may wish to expand to a separate analysis of bipolar 1 and 2 patients. Their response to lithium may be different. Patients with bipolar 2 disorder may not respond equally well to lithium as patients with bipolar 1 disorder.

6) This raises the question whether similar or different inflammatory processes could be at play in bipolar 1 or bipolar 2 disorder. Another question is whether manic and depressed phases elicit similar or different inflammatory processes. These questions should be taken up in the discussion.

7) For better readability, I suggest adding a list of abbreviations in the manuscript showing the abbreviations even in Figure 1.

Author Response

Q1) It remains unclear how the sample was recruited, for instance randomly or consecutively.

R1) We thank the reviewer for the opportunity of clarifying this important point: “Participants were recruited consecutively at the Psychiatric Unit of the University Hospital of Cagliari and of the Department of Medical Sciences and Public Health, University of Cagliari.”

Q2) There are 48/50 individuals in each group. Why were 4 individuals, 2 in each group excluded? How were they statistically handled?

R2) In the BD group two participants were not in a euthymic state. To exclude this additional confounder we included only the euthymic participants. In the control group, one participant was excluded due to missing sociodemographic data, and one due to missing laboratory data. The statistics were thus conducted only with n=48 per group. We added the following sentence: “Indeed, in the BD group two participants were not in a euthymic state. To exclude this possible confounder, we included only the euthymic participants. In the control group one participant was excluded due to missing sociodemographic data and one due to missing laboratory data.”

Q3) Expand on sample size calculation: 5.3 refers to the F value of 5.29 from Kennedy 1996 which was then used by Manchia et al 2019 to investigate differences in melatonin concentrations. The current study however examines KYN/TRP, KYN and QA. How were effect sizes defined?

R3) The study protocol was aimed at assessing differences in levels of melatonin (and of components of its pathway) between patients with BD and healthy controls. We were unable to perform statistical power computation given the absence of previous effect size estimates for the other markers. Thus, we have stressed that this is an exploratory study, which, however, is presenting estimates of magnitude of association that can be used for planning future studies.

Q4) Please explain the Alda scale more clearly, it may be worth to include a text box with an example. Please explain in how far the Alda scale that was designed to measure the treatment effect of lithium can be used for other mood stabilisers or drug combos.

R4) We have revised and explained in more detail the application of the Alda scale and added the literature that validates it as tool for assessing response to other mood stabilizers, and combination therapies. We believe that inserting a text box with the scale would be out of focus for the present paper but reference to the paper of Grof et al. (2002), where the original scale was published, is made in the text.

Q5) In this study, lithium augmented by lamotrigine was more effective than lithium on its own. Please discuss the reasons for that. For figure 2 you may wish to expand to a separate analysis of bipolar 1 and 2 patients. Their response to lithium may be different. Patients with bipolar 2 disorder may not respond equally well to lithium as patients with bipolar 1 disorder.

R5) We agree with the reviewer that also the diagnosis of BD I or II could have influenced the response to treatment and its relationship to the KYN pathway. As only three individuals with BD II were included, the sample is too small to test for differences between these groups. Nonetheless, the two groups are now depicted in different colors in figure 2. Furthermore, we have included this issue as a further limitation of the study and matter of future research due to the limited sample size.

Q6) This raises the question whether similar or different inflammatory processes could be at play in bipolar 1 or bipolar 2 disorder. Another question is whether manic and depressed phases elicit similar or different inflammatory processes. These questions should be taken up in the discussion.

R6) This is a very good point, although the literature on this matter has not yet reach definitive conclusions. It has been repeatedly reported an increase in the levels of proinflammatory cytokines during acute episodes associated with a decrease in neurotrophic support, but how they may vary according to the disease type, disease stage and disease history is still to be clearly demonstrated. There is still debate also concerning the changes in the catabolism of Tryptophan along the serotonin and kynurenine pathways across the disease stages and the contribution of inflammation on these changes. We have included these points in the discussion as suggested.

Q7) For better readability, I suggest adding a list of abbreviations in the manuscript showing the abbreviations even in Figure 1.

R7) A list of abbreviations in the end of the text was added.

Reviewer 4 Report

The manuscript entitled: “Is poor lithium response in individuals with bipolar disorder associated with increased degradation of tryptophan along the kynurenine pathway? “ by Fellendorf et al. (JCM-1621137)  describes  the relationship between lithium response and the measurements of the tryptophan (TRP) to kynurenine (KYN) pathway in 48 patients with bipolar disorder and 48 healthy controls. Due to the numerous corrections in red and blue the whole MS is difficult to follow.

The addional point that should be addressed:

1.Hypothesis is missing.

Author Response

The manuscript entitled: “Is poor lithium response in individuals with bipolar disorder associated with increased degradation of tryptophan along the kynurenine pathway? “ by Fellendorf et al. (JCM-1621137)  describes  the relationship between lithium response and the measurements of the tryptophan (TRP) to kynurenine (KYN) pathway in 48 patients with bipolar disorder and 48 healthy controls. Due to the numerous corrections in red and blue the whole MS is difficult to follow.

  1. R) Thank you for your time considering our manuscript for publication. We used a clear version of the first revision, where now only the changes of the current second revisions are visible.

The addional point that should be addressed:

Q1) Hypothesis is missing.

R1)

Thank you for highlighting this important point. We have now better reported the hypothesis underlying our study. “ We hypothesized that 1) individuals with BD show a greater degradation of TRP along the KYN pathway with increased formation of neurotoxic over neuroprotective metabolites, and 2) the mood-stabilizers lithium and valproate with anti-inflammatory properties are associated with a reduced degradation of TRP along the KYN pathway.”. Please see end of the Introduction.

Round 2

Reviewer 2 Report

 The authors took a few days to implement a number of quick fixes, but the main issues remain. I just think it is not correct to use the current dataset to answer the research questions the authors had in mind. It is misguiding science. 

  • The title remains misguiding as the primary focus was not on lithium, and the findings represent post hoc findings. Only 15 out of 44 patients seem to take lithium, and only 8 do so in mono-therapy (not considering benzodiazepines).
  • Figure 1 already represents the pathway in an interesting way. The positioning of XA is somewhat unfortunate from a visual perspective. It is unclear which molecules are part of the ‘neuroprotective branch’ and the ‘neurotoxic branch’ respectively. Maybe better to work with squares encompassing the metabolites included in the branch.
  • The final paragraph of the introduction remains unclear (why are genetics and gut microbiota mentioned)? What was the research question, what was the hypothesis?
  • My main issues with the paper remain, and are impossible to resolve with the current study design and study sample. The authors aim to investigate the impact of different mood stabilizers on kynurenine metabolites and melatonine. However, sample size is small and all kinds of treatment combinations are being taken by the patients. It is a natural cohort, I’m fully aware, but the dataset is just unfit to investigate the impact of different drugs on the metabolites. It is impossible to tease apart the effects of lithium, valproate, lamotrigine, antipsychotics, benzodiazepines and antidepressants when all patients are taking these agents in a wide number of combinations and dosages. Reporting correlation coefficients in small sample sizes (i.e. 7 and 8 patients) is just bad science. This is misleading, and in order to mention a correlation, a sample size of at least 35-30 is needed, according to multiple statistical handbooks. And in order to do so, a somewhat homogeneous sample (in the light of the primary research question) is needed (so not 8 monotherapeutic lithium patients, and 7 patients with combination therapies).
  • By leaving out analyses for the other mood stabilizers to enforce the post hoc focus on lithium, the reporting on 44 (or 48, depending on the table) patients in the descriptives is a smoke curtain for the fact that primary analyses are only performed on 15 patients. The sample size of this study is 15, not 44 or 48, with two subgroups: 8 mono therapy patients and 7 combination therapy patients. As such, the study is massively underpowered.

Author Response

Q1) The authors took a few days to implement a number of quick fixes, but the main issues remain. I just think it is not correct to use the current dataset to answer the research questions the authors had in mind. It is misguiding science.

R1) We respectfully disagree with the reviewer. This project presents results of an exploratory post-hoc analysis of patients recruited in the context of a translational study exploring the link between the melatonergic system, clinical variables, genomics, and gut microbiota in bipolar disorder. The results we are presenting establish effect sizes of associations and are propaedeutic to future clinical and translational studies.

The argument of misguiding science is an opinion of the reviewer that we respect but do not share.

Q2) The title remains misguiding as the primary focus was not on lithium, and the findings represent post hoc findings. Only 15 out of 44 patients seem to take lithium, and only 8 do so in mono-therapy (not considering benzodiazepines).

R2) The title refers to the main finding of the paper and is appropriate to highlight the most relevant result of the study. The post hoc analysis is acknowledged by the term exploratory.

Q3) Figure 1 already represents the pathway in an interesting way. The positioning of XA is somewhat unfortunate from a visual perspective. It is unclear which molecules are part of the ‘neuroprotective branch’ and the ‘neurotoxic branch’ respectively. Maybe better to work with squares encompassing the metabolites included in the branch.

R3) We thank the reviewer for this point. We have accordingly simplified the scheme of the Kyn pathway highlighting those metabolites object of our analyses.

Q4) The final paragraph of the introduction remains unclear (why are genetics and gut microbiota mentioned)? What was the research question, what was the hypothesis?

R4) The current study is part of a larger trial aimed at investigating in BD the association among genetics, gut-brain axis and biological parameters mostly pertaining to the catabolism of tryptophan along its two main metabolic pathways: 1) the serotonin to melatonin; and 2) the kynurenine. Here, only the laboratory levels of the tryptophan catabolism were analyzed. We agree with you that the presentation is confusing and therefore we deleted the parameters not relevant to this manuscript in the introduction but kept the study description in the methods part. In addition to the description of the research questions, namely the investigation of tryptophan catabolism in BD compared to HC and to assess pilot data about associations of catabolites and mood-stabilizer response, according also the comment of reviewer #4, we added the hypothesis of our study: “We hypothesized that 1) individuals with BD show a greater degradation of TRP along the KYN pathway with increased formation of neurotoxic over neuroprotective metabolites, and 2) the mood-stabilizers lithium and valproate with anti-inflammatory properties are associated with a reduced degradation of TRP along the KYN pathway.”

Q5) My main issues with the paper remain, and are impossible to resolve with the current study design and study sample. The authors aim to investigate the impact of different mood stabilizers on kynurenine metabolites and melatonine. However, sample size is small and all kinds of treatment combinations are being taken by the patients. It is a natural cohort, I’m fully aware, but the dataset is just unfit to investigate the impact of different drugs on the metabolites.

R5) We think that our findings provide elements (such as effect sizes) relevant to the preparation and performance of clinical and translational studies testing the role of the melatonergic system (and of compounds modulating it) in bipolar disorder. The importance of pursuing this goal, although in a preliminary way and with some limitations, has been also acknowledged by the Brain and Behavior Research Foundation through the internationally competitive and prestigious NARSAD Young Investigator award to the PI of this study Dr. Stefano Comai (See funding section).

Q6) It is impossible to tease apart the effects of lithium, valproate, lamotrigine, antipsychotics, benzodiazepines and antidepressants when all patients are taking these agents in a wide number of combinations and dosages. Reporting correlation coefficients in small sample sizes (i.e. 7 and 8 patients) is just bad science. This is misleading, and in order to mention a correlation, a sample size of at least 35-30 is needed, according to multiple statistical handbooks. And in order to do so, a somewhat homogeneous sample (in the light of the primary research question) is needed (so not 8 monotherapeutic lithium patients, and 7 patients with combination therapies).

By leaving out analyses for the other mood stabilizers to enforce the post hoc focus on lithium, the reporting on 44 (or 48, depending on the table) patients in the descriptives is a smoke curtain for the fact that primary analyses are only performed on 15 patients. The sample size of this study is 15, not 44 or 48, with two subgroups: 8 mono therapy patients and 7 combination therapy patients. As such, the study is massively underpowered.

R6) There are several points raised by the reviewer that deserve a comment. The first aspect concerns the sample size and the heterogeneity of treatments. It should be noted that the Alda scale takes into account in B criteria the presence of concomitant treatment such as low dose benzodiazepines when assessing response to treatment. Further, the number of patients treated with lithium and analyzed is indeed 15 but the assessment of response concerns only mono therapy not combination therapy. This has been clarified in the revised version of the paper. Stating that reporting correlations in 15 patients is bad science is not correct since to verify whether a study is underpowered there should be knowledge of the effect size of the reported association. There is no scientific misconduct in reporting results of a correlation of a sample of N = 10 that could be adequately powered if the effect size of the investigated association is of sufficient magnitude. In this study effects sizes are presented, and future studies can be planned with accurate power calculations. In any case, we stressed that this is an exploratory study that could serve future researchers (and our group) to design new studies in the future. However, just to avoid mis-understanding by the readers, we made clearer that this is an exploratory study.

Reviewer 3 Report

The work would benefit from language editing throughout. For instance "Unlike our first hypothesis" instead of "Unlikely our first hypothesis". 

The sentence are very long, which make reading of the manuscript very difficult. For better readability, I suggest that the authors shorten sentences throughout. 

To stress the exploratory nature of this work I suggest to change the wording in the conclusions from "These findings indicate" to "Our preliminary findings suggest"

Author Response

We are again grateful to the reviewer for these insightful comments.

Q1) The work would benefit from language editing throughout. For instance "Unlike our first hypothesis" instead of "Unlikely our first hypothesis". The sentence are very long, which make reading of the manuscript very difficult. For better readability, I suggest that the authors shorten sentences throughout

R1) The manuscript has been extensively revised and changes made according to the reviewer’s indications.

Q2) To stress the exploratory nature of this work I suggest to change the wording in the conclusions from "These findings indicate" to "Our preliminary findings suggest"

R2) Done.

Reviewer 4 Report

I have no additional comments for the authors.

Author Response

We thank the reviewer for the positive assessment of our manuscript.